Task-shifting; supervision; global mental health; mobile phones; implementation

**Corresponding author:**
Noah S. Triplett;
Email: nst7@uw.edu

# Co-developed implementation guidelines to maximize acceptability, feasibility, and usability of mobile phone supervision in Kenya

Noah S. Triplett[1] , Anne Mbwayo[2], Sharon Kiche[1], Lucy Liu[1], Jacinto Silva[1], Rashed AlRasheed[1], Clara Johnson[1], Cyrilla Amanya[3], Sean Munson[4], Bryan J. Weiner[5,6], Pamela Y. Collins[6,7] and Shannon Dorsey[1]

[1]Department of Psychology, University of Washington, Seattle, WA, USA; [2]Department of Psychiatry, University of Nairobi, Nairobi, Kenya; [3]Research Department, Ace Africa Kenya, Bungoma, Kenya; [4]Department of Human Centered Design & Engineering, University of Washington, Seattle, WA, USA; [5]Department of Global Health, University of Washington, Seattle, WA, USA; [6]Department of Health Services, School of Public Health, University of Washington, Seattle, WA, USA and [7]Department of Psychiatry and Behavioral Sciences, University of Washington, Seattle, WA, USA

## Abstract

Opportunities exist to leverage mobile phones to replace or supplement in-person supervision of lay counselors. However, contextual variables, such as network connectivity and provider preferences, must be considered. Using an iterative and mixed methods approach, we co-developed implementation guidelines to support the implementation of mobile phone supervision with lay counselors and supervisors delivering a culturally adapted trauma-focused cognitive behavioral therapy in Western Kenya. Guidelines were shared and discussed with lay counselors in educational outreach visits led by supervisors. We evaluated the impact of guidelines and outreach on the acceptability, feasibility, and usability of mobile phone supervision. Guidelines were associated with significant improvements in acceptability and usability of mobile phone supervision. There was no evidence of a significant difference in feasibility. Qualitative interviews with lay counselors and supervisors contextualized how guidelines impacted acceptability and feasibility – by setting expectations for mobile phone supervision, emphasizing importance, increasing comfort, and sharing strategies to improve mobile phone supervision. Introducing and discussing co-developed implementation guidelines significantly improved the acceptability and usability of mobile phone supervision. This approach may provide a flexible and scalable model to address challenges with implementing evidence-based practices and implementation strategies in lower-resourced areas.

## Impact statement

Task-shifting, in which lay counselors without formal mental health training are trained and supported to deliver mental health interventions, is an acceptable and effective way of delivering mental health treatment. However, there are challenges with ensuring the feasibility and sustainability of supervision in task-shifting, particularly in areas with fewer numbers of trained supervisors. Mobile phones may present an opportunity to support lay counselors from afar, but special attention must be paid to ensure that lay counselors are being appropriately supported via mobile phones. Additionally, given the vast differences between mobile phone access, cellular network connection, and other contextual variables in many low-resource areas, approaches to support mobile phone supervision must be flexible and encourage tailoring to specific contexts. The current research describes the co-development of a flexible and pragmatic approach to support mobile phone supervision with lay counselors in Western Kenya. Lay counselors and supervisors delivering a culturally adapted trauma-focused cognitive behavioral therapy participated in a co-design process to design an approach to support mobile phone supervision. The co-design process resulted in the development of implementation guidelines, which were to be shared and discussed in a brief educational outreach visit between supervisors and lay counselors. We compared the effects of the guidelines and visit on the acceptability, feasibility, and usability of mobile phone supervision between lay counselors who did and did not receive guidelines. The introduction and discussion of guidelines was associated with improvements in the acceptability and usability of mobile phone supervision. In comparing our approach and benefits to other implementation strategies, we highlight the importance of scalability, flexibility, and simplicity in our approach. We also note the importance of valuing the expertise of lay counselors and supervisors to select and tailor implementation strategies.

## Background

Though there are mental health treatment gaps in nearly every country, there are larger treatment gaps in low-to-middle-income countries (LMICs), compared to high income countries, due to fewer trained mental health care providers (Demyttenaere et al., 2004; Kohn et al., 2004; Moitra et al., 2022). Task-shifting has emerged as a potential strategy to address the human resource shortages that, in part, contribute to the mental health treatment gap (Murray et al., 2014; Weiss et al., 2015; Chibanda et al., 2016; Hoeft et al., 2018). As part of task-shifting models, lay counselors (e.g., teachers and health workers) without formal mental health training are trained and supported to deliver interventions (van Ginneken et al., 2013). Evidence supports the effectiveness of task-shifting to deliver evidence-based practices (EBPs) for mental, neurological, and substance use disorders in LMIC (van Ginneken et al., 2013; Murray et al., 2014; Weiss et al., 2015; Chibanda et al., 2016; Hoeft et al., 2018; Dorsey et al., 2020b). As task-shifting continues to expand, more research is needed to understand how to fully embed and sustain task-shifting in LMIC, including how to sustainably supervise and support lay counselors (Padmanathan and De Silva, 2013).

A growing body of literature has examined how technology can be used to build capacity and scale-up task-shifting globally (Naslund et al., 2019), including as a tool to support supervision during in-person meetings (Rahman et al., 2019). In particular, mobile phones may offer a cost-effective and accessible means of replacing or supplementing in-person supervision during task-shifting; however, it is important to consider and address the various challenges and nuances that may accompany mobile phone supervision in LMIC (Triplett et al., 2023). Though less research has examined mobile phone and digital technology use in task-shifting, a large body of research has examined the challenges and opportunities of mobile phone use in LMIC (Kusimba et al., 2015; Murphy and Priebe, 2017; Wyche and Olson, 2018). Research has noted the importance of considering the limitations of mobile phones when designing and implementing digital technologies in LMIC (Wyche and Murphy, 2012), including digital supports to healthcare in LMICs (Henry et al., 2016; Feroz et al., 2020).

Given the challenges with implementing digital technologies in LMIC, flexible and adaptable implementation strategies are needed. Guidelines that contain recommendations to guide mobile phone supervision may be one pragmatic strategy; however, specific attention must be paid to ensure guidelines are flexible, account for contextual differences, and are appropriately introduced to clinicians. Guidelines are frequently used to support implementation of healthcare innovations, including within LMIC (Francke et al., 2008; Nabyonga Orem et al., 2012). Despite this, there are noted challenges with dissemination of guidelines, such as guidelines being too difficult to understand, challenges implementing without support from peers and managers, and insufficient staff and time to implement guidelines (Francke et al., 2008). These challenges may be intensified in lower-resource settings, such as LMIC (Nabyonga Orem et al., 2012). There is evidence to suggest that, with proper planning, stakeholder engagement and dissemination, guideline implementation can result in desired impacts on clinician behavior (Peters et al., 2022). Thus, if guidelines are designed to allow for flexibility and tailoring and are combined with other minimally intensive implementation strategies, such as educational outreach visits to discuss and plan for their implementation, they may be an appropriate and resource-efficient means of supporting mobile phone supervision.

The present article outlines the results of an iterative, mixed methods study that evaluated the effects of sharing co-developed implementation guidelines in a brief educational outreach visit on the acceptability, feasibility, and usability of mobile phone supervision for lay counselors in Kenya. Guided by human-centered design (HCD), we engaged supervisors and lay counselors to co-develop the implementation guidelines and determine the most feasible way of discussing and disseminating the guidelines – educational outreach visits. Leveraging an ongoing stepped wedge trial evaluating the implementation of trauma-focused cognitive behavioral therapy (TF-CBT; see Dorsey et al., 2020a), we compared effects of guidelines and outreach across quantitative measures of acceptability, feasibility, and usability. Results were contextualized with semi-structured interviews to explain the benefits of guidelines and outreach as well as how they might have impacted outcomes.

## Methods

### Study design

This study used an iterative and mixed methods approach, with qualitative interviews and a HCD workshop informing the development of the implementation guidelines and outreach strategy, which was subsequently tested in a randomized pilot trial. Full trial procedures are described elsewhere (Triplett et al., 2021). The trial was situated within a larger trial that examined the effectiveness and implementation of a locally adapted version of TF-CBT (Cohen et al., 2006) in Bungoma, Kenya (Building and Sustaining Interventions for Children [BASIC]; see Dorsey et al., 2020a for parent trial protocol). BASIC utilizes an eight-session version of TF-CBT ("*Pamoja Tunaweza*"), which was adapted by longstanding Kenyan partners at Ace Africa (i.e., supervisors and counselors) for cultural relevance and acceptability. Lay counselors work together in groups of three to provide the treatment in a group-based format and are trained and supervised by three Kenya-based supervisors. Lay counselors work within two governmental sectors in Kenya, identified as potentially viable systems for scale-up – Education (via teacher delivery) and Health (via community health volunteer [CHV] delivery). Supervisors are Ace Africa employees who were previously trained and subsequently delivered the treatment in a randomized controlled trial (Dorsey et al., 2020b) that preceded the current trial.

To develop and evaluate the guidelines and outreach, we made use of the parent study's stepped-wedge cluster-randomized design. BASIC was comprised of seven sequences, in which a total of 40 schools and 40 communities surrounding the schools were randomized to begin TF-CBT implementation at seven different time points throughout the study. Counselors from sequences one through five of the BASIC trial were randomly selected to participate in qualitative interviews regarding their experiences using mobile phones for clinical supervision, which included strategies to improve use (Triplett et al., 2023). All BASIC supervisors also participated in interviews regarding their experiences using mobile phones to provide supervision.

Following interviews and thematic analysis, BASIC trial supervisors presented back results from the semi-structured interviews and led discussion with lay counselors on any additions, edits, or clarifications they felt were needed for the themes. This workshop also included other HCD activities, which aimed to engage participants and determine how interview findings could be translated into actionable solutions for counselors for the implementation guidelines. Following the workshop, the supervisors

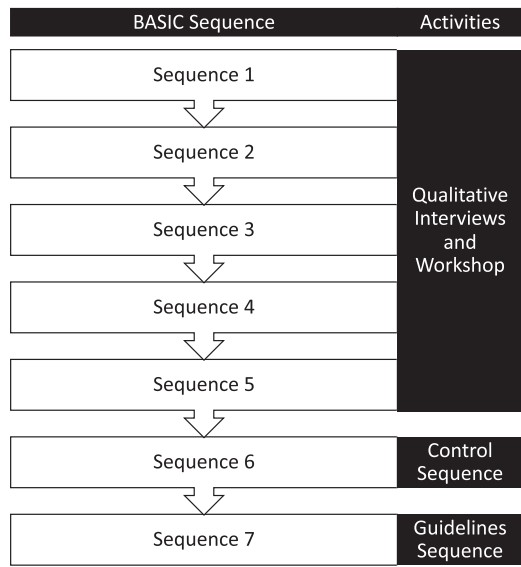

**Figure 1.** Trial sequence and activities.

**Table 1.** Sample implementation guidelines for Goal A

| |
| --- |
| *Goal A*: Explain to counselors that we will use in-person and mobile phone supervision together to ensure they are supported through PT delivery. |
| A1. We will provide in-person supervision at least four times during PT delivery. This will be at the beginning, middle, and end of PT groups. |
| • Supervision will be provided through mobile phones throughout the PT program. Counselors should feel free to use their mobile phones to communicate with their supervisors for support at any point during the program. |
| A2. If you need additional support, you may request that your supervisor comes for additional in-person supervision visits. |
| • Other counselors have requested support when facing challenges with certain topics or clarifying the goals of the PT program. |

and principal investigator of the project (NST) reviewed all solutions and compiled them into a one-page document for supervisors and lay counselors to review together (the implementation guidelines). The co-development process occurred while sequence six of the BASIC trial was implementing TF-CBT, and they served as a no-guidelines control group. After the guidelines were developed, they were implemented with all counselors in sequence seven. This process is depicted in Figure 1. The Institutional Review Boards (IRB) at the University of Washington and Kenya Medical Research Institute approved all study procedures.

### Implementation guidelines

The guidelines are presented in Supplementary Material S1. One example goal and corresponding strategies are also presented in Table 1. As described above, the guidelines included multiple strategies that were clustered under the four goals: A) Explain to counselors that they will use in-person and mobile phone supervision together to ensure counselors are supported through delivery. B) Ensure counselors get all the information and support that they need through mobile phone supervision. C) Plan for any challenges with network connection. D) Decrease distractions and disruptions during mobile phone supervision. All strategies were presented as optional, and counselors were encouraged to select and attempt strategies that they felt best matched their unique school and community contexts. Counselors were not required to implement any solutions, and there was no expectation of continued follow-up from supervisors.

### Participants

Participants included lay counselors recently trained in and who subsequently delivered TF-CBT as part of the parent trial ($N = 59$; 29 teachers; 30 CHVs). All parent trial supervisors ($N = 3$) also participated in semi-structured interviews. Lay counselors and supervisors were delivering TF-CBT as part of sequences six and seven of the BASIC parent trial. To be eligible to participate in the BASIC trial, lay counselors had to be nominated from leaders at their site (i.e., head teacher and/or deputy teacher, or community health extension worker). Nominations were based on criteria to identify the most appropriate counselors (e.g., desire to be counselors; experience working with children). There were no exclusion criteria. All participants provided informed consent at the time of enrollment.

### Procedure

We compared acceptability, feasibility, and usability measures of mobile phone supervision from counselors in sequences six and seven of the parent trial. Lay counselors in sequence six ($n = 29$) did not receive any additional support or guidelines for mobile phone supervision. Lay counselors in sequence seven ($n = 30$) received the implementation guidelines document. Supervisors reviewed and discussed the guidelines with sequence seven lay counselors in an educational outreach visit before they began TF-CBT delivery. These visits were integrated into the first in-person supervision meeting for the parent trial, which occurred shortly following training and before lay counselors began TF-CBT groups. The addition of mobile phone supervision outreach added approximately 30 min to the initial in-person supervision meeting. Supervisors traveled to lay counselors' sites and facilitated separate meetings with each group of lay counselors (i.e., the meetings included the supervisor and the three counselors who co-delivered the treatment in each site). Some phone contact occurred prior to this meeting to plan a meeting time, then phone supervision continued following this meeting.

Per the parent trial protocol, in-person supervision happened at least four times during each TF-CBT group (8 weeks). Through both sequences, supervisors were also available to conduct additional in-person supervision as needed with lay counselors. The manipulation was solely the introduction of guidelines and an educational outreach visit to support successful supervision by mobile phone. Lay counselors in both sequences completed measures of mobile phone supervision acceptability and feasibility, as well as a measure of usability, after delivering two rounds of TF-CBT. All three supervisors and a randomly selected sub-sample of lay counselors from sequence seven ($N = 12$; six teachers; six CHVs) also participated in qualitative interviews to gather more information on their experience with the guidelines and perspectives on mobile phone supervision. The final interview guide is included in Supplementary Material S2.

### Measures

Measures were adapted from existing measures, prioritizing acceptability and feasibility measures already translated and used cross-culturally in the parent trial (i.e., BASIC) and other studies globally.

All adaptations to the usability measure were made following established procedures to ensure common understanding of the construct (Dorsey et al., 2020a) and completed in consultation with longstanding Kenyan partners on the trial.

### Acceptability

The four-item Acceptability of Intervention measure (Weiner et al., 2017) was adapted and used to assess lay counselor perspectives of mobile phone supervision acceptability (e.g., "I like mobile phone supervision"). Scores range from 1 to 5, with higher scores representing greater acceptability. This brief, pragmatic measure has acceptable internal consistency and test–retest reliability in other samples (Weiner et al., 2017), and had good internal consistency in our sample (α = 0.89).

### Feasibility

The four-item Feasibility of Intervention measure (Weiner et al., 2017) was used to assess lay counselor perspectives of mobile phone supervision feasibility (e.g., "mobile phone supervision seems doable in this school/community"). Scores range from 1 to 5, with higher scores representing greater feasibility. This measure has acceptable internal consistency and test–retest reliability in other samples (Weiner et al., 2017), and had excellent internal consistency in our sample (α = 0.93).

### Usability

The 10-item Intervention Usability Scale (IUS; Lyon et al., 2020) was used to assess lay counselor perspectives of mobile phone supervision usability (e.g., "mobile phone supervision was easy to use"). Scores range from 0 to 100, with higher scores representing greater usability. The IUS has acceptable internal consistency in other samples (Lyon et al., 2020), and had good internal consistency in our sample (α = 0.87).

### Analysis

We present descriptive statistics (mean, standard deviation, and range) to understand counselor and supervisor ratings of acceptability, feasibility, and usability following mobile phone supervision. We also conducted independent samples *t*-tests to compare average ratings of acceptability, feasibility, and usability across sequences that did and did not receive the guidelines and educational outreach visit. Following best practices for smaller sample sizes (Weissgerber et al., 2016), quantitative data are also visualized to illustrate any outliers and differences between sequences. All quantitative analyses were conducted using R (R Core Team, 2022).

Recordings from interviews were transcribed and identifying information was removed. Transcripts were coded in Dedoose (QSR International Pty Ltd., 2018) by researchers in the US and Kenya. Analysis was informed by thematic analysis (Braun and Clarke, 2006). Kiswahili interviews were translated by native Kiswahili speakers and trained translators. To develop an initial codebook, coders reviewed three transcripts independently, then met to identify potential codes and produce an initial codebook. This codebook was subsequently applied and refined on the remaining interviews. All coding was done independently, and consensus was reached through group dialog (Hill et al., 1997). We followed a QUAN → qual mixed methods approach for data explanation, using the embedded qualitative data to elaborate on or contextualize quantitative results (Palinkas et al., 2011).

## Results

### Sample demographics and quantitative results

Our sample included 29 teachers and 30 CHVs who delivered TF-CBT in sequences six and seven of the BASIC trial. Demographics are presented in Table 2. Differences in mobile phone supervision acceptability, feasibility, and usability between sequence six and sequence seven are presented in Table 3 and Figure 2. As shown, there were statistically significant differences in acceptability and usability between counselors who did receive guidelines and those who did not. Given potential conceptual overlap between

**Table 2.** Demographics and baseline characteristics

| Characteristic | No guidelines sequence six (n = 29) No. (%) | Guidelines sequence seven (n = 30) No. (%) |
|---|---|---|
| Sector (counselor type) | | |
| Education (teachers) | 14 (48.3) | 15 (50.0) |
| Health (CHVs) | 15 (51.7) | 15 (50.0) |
| Sex | | |
| Male | 11 (37.9) | 12 (40.0) |
| Female | 18 (62.1) | 18 (60.0) |
| Highest level of education | | |
| Primary education | 7 (24.1) | 2 (6.7) |
| Secondary education | 8 (27.6) | 13 (43.3) |
| Certificate | 2 (6.9) | 9 (30.0) |
| Diploma certificate | 8 (27.6) | 4 (13.3) |
| Master's degree | 4 (13.8) | 2 (6.7) |
| Received prior training in psychosocial counseling | | |
| No | 14 (48.3) | 20 (66.7) |
| Yes | 15 (51.7) | 10 (33.3) |
| Provided prior psychosocial counseling | | |
| No | 6 (20.7) | 11 (36.7) |
| Yes | 23 (79.3) | 19 (63.3) |
| Experience working with children/adolescents | | |
| No | 7 (24.1) | 6 (20.0) |
| Yes | 22 (75.9) | 24 (80.0) |
| Experience working with parents/guardians | | |
| No | 5 (17.2) | 10 (33.3) |
| Yes | 24 (82.8) | 20 (66.7) |
| | *M (SD)* | *M (SD)* |
| Age (in years) | 42.9 (7.3) | 40.9 (6.7) |
| Years of part-time psychosocial counseling experience | 6.2 (3.7)[a] | 8.7 (8.1)[b] |
| Years of full-time psychosocial counseling experience | 0.04 (0.2)[a] | 5.4 (9.3)[b] |

[a]*n* = 23.
[b]*n* = 19.

quantitative measures, we also present Pearson correlation coefficients between outcome measures in Supplementary Material S3.

## Qualitative results

Lay counselors and supervisors in sequence seven described benefits of the mobile phone supervision guidelines, and specifically the benefits of having a dedicated educational outreach visit to discuss strategies to improve mobile phone supervision prior to its implementation. The qualitative themes on benefits of the guidelines and outreach included: setting expectations for mobile phone supervision; emphasizing the importance of mobile phone supervision; increasing comfort with mobile phone supervision; and, sharing strategies for mobile phone supervision.

### Setting expectations for mobile phone supervision
Guided by the guidelines document (see Goal A), supervisors discussed what counselors might expect during mobile phone supervision and worked to collaboratively set expectations for

how supervisors and counselors themselves would approach mobile phone supervision, including expectations for carrying phones and ensuring their phones were working. As one counselor described, "this meeting revealed to me… that everyone is required (to have a phone), if you do not have a phone, try your best to have a phone so that any info comes from a supervisor (you won't miss it)…." Counselors and supervisors also discussed how the educational outreach visit enabled them to set expectations for scheduling and rescheduling mobile phone supervision, which ultimately led to more successful mobile phone supervision meetings: "The other times before we did these meetings of mobile phone supervision, we were just calling (counselors)…. But now due to these meetings, it made communication work better because now we were planning earlier before the calls."

### Emphasizing the importance of mobile phone supervision
As one supervisor described, having educational outreach visits and discussions about mobile phone supervision before beginning treatment delivery "let the counselors know that it's not all about in-person (supervision). Because initially, (counselors) were thinking that supervision is mainly important if it's in person, but then they took the phone supervision more seriously." Counselors also noted how the visit and guidelines influenced their perceptions of the importance of mobile phone supervision, with one counselor explaining that "(they) had never been subjected to telephone supervision (before this program)…. But, this time (their) supervisor was able to teach (them) the importance of having telephone supervision…." Defining the importance of mobile phone supervision was closely related to setting expectations for supervision, as counselors and supervisors both noted that setting expectations for

**Table 3.** *t*-Test results for acceptability, feasibility, and usability

| | No guidelines (sequence six) | | Guidelines (sequence seven) | | | | |
|---|---|---|---|---|---|---|---|
| | *M* | *SD* | *M* | *SD* | *t*-Value | *df* | *p* |
| Acceptability | 4.08 | 0.34 | 4.35 | 0.18 | −2.05 | 57 | 0.04 |
| Feasibility | 3.78 | 0.79 | 4.15 | 0.29 | −1.92 | 57 | 0.06 |
| Usability | 66.12 | 18.46 | 75.58 | 12.05 | −2.34 | 57 | 0.02 |

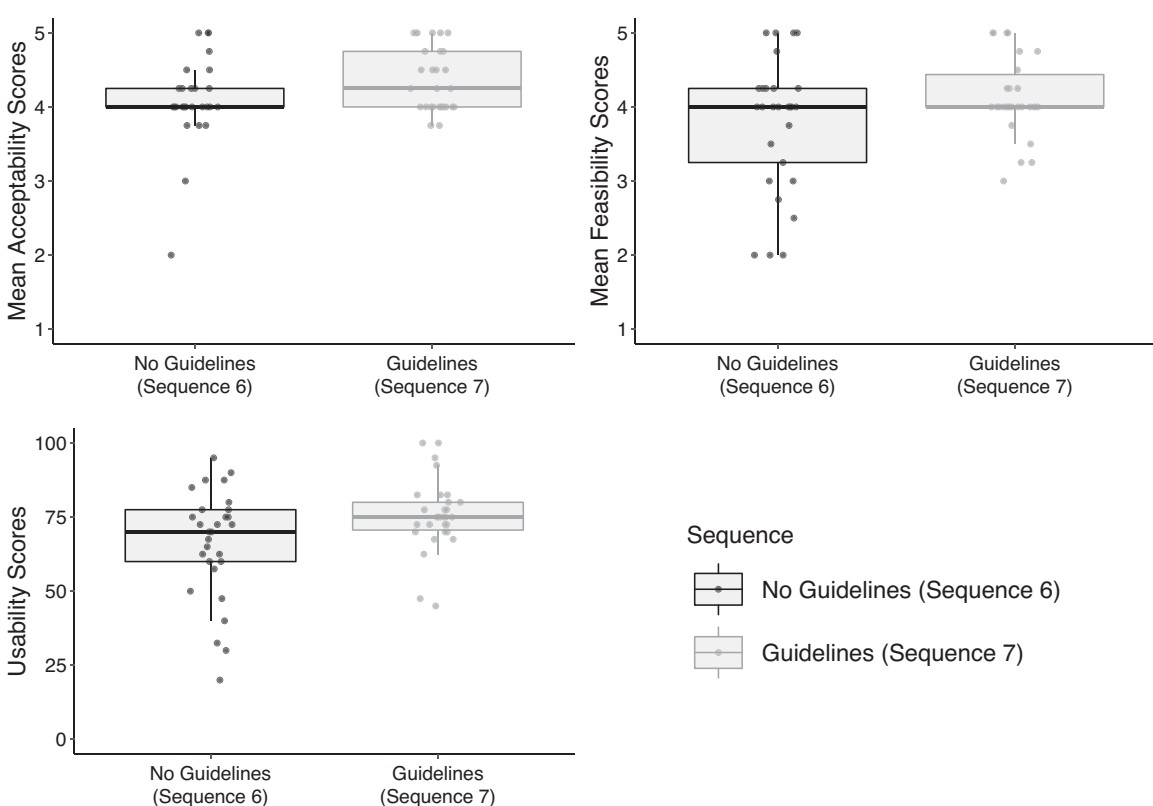

**Figure 2.** Differences in outcomes by guidelines condition.

carrying phones and being available for phone supervision meetings reinforced the importance of mobile phone supervision.

### Increasing comfort with mobile phone supervision

This was facilitated by a variety of factors, including decreasing counselor anxiety surrounding mobile phone supervision and increasing comfort with specific strategies. Counselors reported feeling some anxiety calling and getting something wrong or inconveniencing their supervisors; however, as one counselor reported, "(the meeting) put me in a good position to talk on the phone, getting used to doing something like a telephone interview without trembling." This increased comfort was essential for counselors to engage in all the mobile phone supervision strategies, including those that were intended to reduce counselor airtime usage, such as flashing and reverse calling. Flashing required lay counselors to call and quickly hang up such that their supervisor's airtime was charged as opposed to their own. Reverse calling is a specific way of dialing that charges the recipient's airtime. As one supervisor noted, "we made them comfortable to flash us initially. Initially, (the counselors) were like, sometimes they didn't have their time, so I made the counselors comfortable to just flash me."

### Sharing strategies for mobile phone supervision

The experience of receiving mobile phone supervision was new for counselors, who noted the benefits of not only learning to use phones but also strategies to make their use easier: "We were taught on how we were to use the phones. Now these are the strategies on how we were to use (them)." As one supervisor explained, the educational outreach visit was a more comprehensive and efficient way of sharing strategies that might have naturally occurred later: "Without the meeting, we could have done some of the activities, like making reminder phone calls, but it could have not created as much emphasis, and they could have not gotten as much more information as they did during the meetings."

### Goals and strategies

In describing the benefit of sharing strategies for mobile phone supervision, counselors and supervisors referenced strategies that aligned with each guidelines document goal.

### Goal A: Explain logistics of mobile phone supervision

Among the most frequently mentioned (relative to the overall goal) were requests for in-person supervision, in which counselors noted that having this option was essential in building their confidence and supporting them in handling clinical challenges: "As I was delivering the lesson, the girls were so emotional…. It was even difficult for me to continue with the lesson. So, I called my supervisor…." This flexibility and additional support were crucial for supporting counselors as they assumed new counseling roles and the associated emotional difficulties.

### Goal B: Ensure counselors get all needed information and support

Counselors frequently mentioned the importance of sharing contact information at the educational outreach visit, both with supervisors and co-counselors. They also discussed how they would vary their communications between phone calls, SMS, and WhatsApp messages. One strategy that was stressed to ensure all counselors were receiving information was to communicate messages to the entire group, either via similar yet separate messages, singular messages to the entire group, or even group calls (i.e., conference calls or calling one counselor and having them place their phone on speaker). Counselors discussed how this helped them to not miss important information from their supervisors, "We could not worry because at least the three of us, there is no way we could (all miss messages)." Another counselor noted that group communications also had the benefit of facilitating learning between counselors: "I learn more because in the, that is WhatsApp, we are in a group, then we share. And as you are sharing that with our supervisor, you get more information from other co-teachers."

A crucial piece of ensuring counselors received all needed information and support was ensuring that their phones were supplied with airtime, charged, and properly working. To address airtime shortages, counselors discussed various strategies to communicate when their airtime was low, such as "reverse calling" (i.e., calling and charging the recipient's airtime), "flashing" (i.e., calling and hanging up quickly such that the other person returns the call and charges their own airtime), and sending a free "please call me" text message to supervisors. As one counselor described, "So, we were told in the meeting that in case we don't have airtime, you are just supposed to flash or even use WhatsApp. We have please call, send a please call me. And our supervisor will call us back." Counselors also referenced strategies to ensure their phones remained charged and minimize battery use.

### Goal C: Plan for challenges with network connection

Counselors mentioned the importance of updating their phones and SIM cards to improve their network connection. Counselors also indicated that the educational outreach visit was helpful in facilitating their identifying locations with strong network connection where they could take mobile phone supervision calls and receive messages. One counselor noted, "I also learned that I have to find a good place where the network is stable to talk to the supervisor for things to be better…." For some counselors, having scheduled mobile phone supervision meetings (another strategy) was also important to ensure they could be in a location with network connection: "there's just a particular point where we have the network…. So, (the guidelines) forced us to make sure that at a certain time, you need to move somewhere at a certain point so that you get some communication."

### Goal D: Decrease distractions and disruptions

Given that counselors frequently completed supervision meetings at the schools in which they delivered treatment, they noted the importance of identifying quiet and secure places to have supervision phone calls: "during the meeting (the supervisor) also told us to make sure that we are confidential… to make sure that when we decide that we have to make a call, we were to look for a quiet place." Advance notice of when supervision would occur was crucial for counselors to go to these locations on the school ground, prepare themselves, and notify others that they would be busy with mobile phone supervision. One counselor noted, "I can also tell those who are around, who may make noise, that I have (supervision) now, let us not be noisy. So that when we start, we do not get any disturbance…."

## Discussion

Co-developed implementation guidelines and educational outreach visits were associated with improved acceptability and usability of mobile phone supervision. Qualitative interviews with lay counselors and supervisors contextualized how the guidelines and visits impacted acceptability and usability – by setting expectations for

mobile phone supervision, emphasizing its importance, increasing comfort, and sharing strategies to improve mobile phone supervision. In discussing how specific strategies were utilized across different contexts, interviews showed how the guidelines were implemented across contexts and highlight the flexibility of our approach. Our results indicate that, with some flexibility and minimal support around their introduction and dissemination, guidelines may be a viable tool to implement and sustain some clinical innovations or implementation strategies across areas with varying levels of resources.

### Flexible implementation guidelines

Given the documented challenges with implementing and sustaining interventions and their components (e.g., mobile phone supervision), there have been calls for greater research on how to best tailor implementation support (Baker et al., 2015; Powell et al., 2017; Waltz et al., 2019). Research has examined implementation support approaches that identify and match implementation strategies to address specific determinants prior to implementation (Baker et al., 2015) as well as more intensive approaches to provide individualized, on-going implementation support, such as implementation facilitation (Smith et al., 2022) and technical assistance (Katz and Wandersman, 2016). Though effective, there may be challenges with implementing or scaling complex, multi-faceted implementation strategies in lower-resource settings. We expand the existing research by examining the effect of a pragmatic multi-faceted implementation strategy: co-developed implementation guidelines for mobile phone supervision and brief educational outreach visits. Incorporated into our guidelines were multiple suggested strategies for improving mobile phone supervision, thereby providing guidance to counselors while accounting for needed flexibility and tailoring to different contexts (e.g., multiple options for communicating with different levels of network bandwidth).

In comparing our approach and benefits to other implementation strategies, we highlight the importance of scalability, flexibility, and simplicity in our approach. Noting resource constraints and the need for scalability, we were guided by the concept of "minimal intervention needed to produce change" (Glasgow et al., 2014). Though formative work was completed to develop the implementation guidelines (Triplett et al., 2023), the educational outreach visits were fully integrated into existing research and clinical activities. Lay counselors were supplied with the paper guidelines documents, but they were not required to track strategies or evaluate their own fidelity to the strategies. Increasingly, implementation researchers are trying to understand how little support or intervention can be provided to achieve desired impacts (Glasgow et al., 2014; Lyon et al., 2022). Our approach was designed with this in mind, prioritizing pragmatism and scalability with minimal resources.

Our approach also prioritized flexibility and acknowledged the expertise of lay counselors and supervisors. Children's mental health implementation research has often only consulted with stakeholders to understand challenges following implementation efforts, which limits the ability of researchers to empower stakeholders, preempt challenges to EBP implementation, and ensure implementation success (Triplett et al., 2022). Conversely, our approach enabled lay counselors and stakeholders to develop their own guidelines. It also allowed for flexibility and daily tailoring within guidelines by lay counselors and supervisors. As interview quotations illustrate, this approach was highly acceptable as it increased counselor comfort enabled them to seek and receive more support while implementing the intervention. Much of the existing literature on acceptability has focused on the acceptability of interventions to stakeholders (Lewis et al., 2015). However, it is also important to investigate the acceptability of implementation strategies (Proctor et al., 2013), particularly those that may be critical to implementation and eventual sustainment, such as mobile phone supervision. Particularly given the power dynamics inherent in both global mental health and implementation science research, where researchers often put forward EBPs to implement, it is important to assess and potentially improve the acceptability of interventions and implementation strategies.

### Impacts on mobile phone supervision

Our approach was associated with greater acceptability and usability of mobile phone supervision. There is no evidence that the introduction of guidelines and the educational outreach visits were associated with significant changes in mobile phone supervision feasibility. There are significant challenges in implementing digital health solutions in LMIC (Kusimba et al., 2015; Murphy and Priebe, 2017; Wyche and Olson, 2018), including with lay counselors (Triplett et al., 2023). By focusing specifically on these challenges within our guidelines, we may have even increased the salience of barriers to mobile phone supervision. Interestingly, though average feasibility ratings were high, there was great variability in lay counselor reports of feasibility across both sequences, meaning counselors varied more in their perception of mobile phone supervision as compared to acceptability or usability. This again highlights the importance of flexible approaches that allow tailoring to individual contexts, given that initial perceptions of feasibility may be discrepant across contexts.

The guidelines most often included short-term solutions to challenges that could be implemented with minimal resources. Other potentially viable strategies to improve feasibility may have included the provision of additional resources (e.g., phones or airtime) or formal workload adjustment to allow for lay counselors to receive supervision. There was no intervention on structural challenges (e.g., network coverage), which gave rise to the barriers and could ultimately impact feasibility. Instead, as evidenced in the qualitative findings, the guidelines' focus on individual actions supported counselors through challenges. Though these types of solutions may be necessary in the present, it should be stressed that equitable and sustainable implementation of mental health interventions, particularly when projects are driven by U.S. investment in lower-resource areas, must also aim to address the structural and systemic factors that have created and maintain inequities (e.g., working to understand and address global power structures that dictate resource allocation).

### Limitations

Our implementation guidelines were co-developed alongside lay counselors and supervisors from previous sequences in the trial and were very specific mobile phone supervision in Kenya. While we believe the flexibility and rationale underlying the approach would extrapolate well to other projects, further research is needed to evaluate similar approaches across contexts. Similarly, though the cluster-randomized trail design may have protected against some threats to validity in terms of participant selection, our design cannot account for natural improvements in mobile phone supervision that might have occurred due to the progression of time. Additionally, because of the aims of this trial and desire to reduce reporting burden on participating counselors, we did not track the

impact of the facilitation on frequency of mobile phone contacts or the frequency with which specific strategies were used. Further research is needed to investigate the impacts of the program on these as well as other implementation outcomes (i.e., what are the effects of mobile phone supervision on intervention fidelity and clinical outcomes). Finally, our coding team consisted of one native Kiswahili speaker who consulted the Kiswahili audio and answered team questions related to translation and coding for Kiswahili interviews. However, it is possible that some nuance in conversations was lost in the translation to English.

## Conclusion

Mobile phones may present an opportunity to increase access to lay counselor supervision; however, implementation of mobile phone supervision must acknowledge and address contextual barriers. Our approach significantly impacted the acceptability and usability of mobile phone supervision. Qualitative interviews with lay counselors and supervisors contextualized how the guidelines and educational outreach visits impacted acceptability and usability – by setting expectations for mobile phone supervision, emphasizing importance, increasing comfort, and sharing strategies to improve mobile phone supervision. Together, results highlight the importance of minimally intensive and flexible implementation supports that can be tailored across contexts and empower stakeholders to select and implement their own solutions. Importantly, we argue this must not come at the cost of addressing structural issues that give rise to downstream barriers. Both elements are crucial to ensuring interventions can be implemented and sustained across areas with varying levels of resources.

**Open peer review.** To view the open peer review materials for this article, please visit http://doi.org/10.1017/gmh.2023.23.

**Supplementary material.** The supplementary material for this article can be found at http://doi.org/10.1017/gmh.2023.23.

**Data availability statement.** The datasets generated and analyzed for this study are available from the corresponding author upon reasonable request.

**Acknowledgments.** We thank the Ace Africa TF-CBT supervisors, community mobilizer, and interviewers: Elijah Agala, Moses Malaba, Emmanuel Muli, Bernard Nabalia, Micah Nalianya, Michael Nangila, Omariba Nyaboke, Daisy Okoth, Annette Sulungai, Sylvia Wafula, and Nelly Wandera. Their hard work, dedication, and high integrity made this research possible. We are also grateful to the Kenyan Ministry of Health, Ministry of Education, Teachers Service Commission, participating schools, communities, children, and families who are taking part in this trial and the parent trial.

**Author contribution.** Conceptualization: N.S.T., S.M., B.W., P.C., S.D.; Data curation: N.S.T.; Formal analysis: N.S.T., S.K., L.L., J.S.; Funding acquisition: N.S.T.; Project administration: C.A.; Supervision: N.S.T., A.M., S.M., B.W., P.C., S.D.; Writing – original draft: N.S.T., S.K., L.L., J.S.; Writing – review and editing: N.S.T., A.M., S.K., L.L., J.S., R.A., C.J., C.A., S.M., B.W., P.C., S.D.

**Financial support.** Funding for this study has been provided by the National Institute of Mental Health (NIMH) through a National Research Service Award (NRSA) Individual Pre-Doctoral Fellowship (NIMH F31 MH124328; N.S.T., PI) and the larger randomized clinical trial that is providing the sample of lay counselors and supervisors (NIMH R01 MH112633; Whetten and S.D., MPI). Funding was also provided by a dissertation grant from the Health Policy Research Scholars Program, a program of the Robert Wood Johnson Foundation. S.M.'s time was supported through the University of Washington ALACRITY Center which is funded through the National Institutes of Health (P50 MH115837). The funders/sponsor have no role in study design or analysis.

**Competing interest.** N.S.T. has received honoraria for speaking engagements with the Robert Wood Johnson Foundation, a funder of this work. S.D. has received grants and honorariums for providing training and consultation on the treatment model that was adapted and delivered by the lay counselors (trauma-focused cognitive behavioral therapy). The remaining authors declare that the research was conducted in the absence of any significant commercial or financial relationships that could be construed as a potential conflict of interest.

**Ethics standard.** All authors declare to adhere to the publishing ethics of Global Mental Health.

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
