## [Reviewer Report]

To Drs. Bass and Chibanda,

Editors-in-Chief 

Cambridge Prisms: Global Mental Health

My co-authors and I are submitting the attached article titled, Co-developed implementation guidelines to maximize acceptability, feasibility, and usability of mobile phone supervision in Kenya, for your consideration for publication in Cambridge Prisms: Global Mental Health. Little scholarship has focused how to scale up and sustain task-shifting of evidence-based practices (EBPs) for mental health disorders in lower-resourced settings, including how to sustainably supervise lay counselors. We describe methods and results from an iterative, mixed methods trial that engaged lay counselors and supervisors providing trauma-focused cognitive behavioral therapy (TF-CBT) in Kenya to redesign supervision to most effectively leverage mobile technology. 

We briefly outline the importance of developing scalable and sustainable approaches to supervision—an essential yet resource-intensive implementation strategy for ensuring fidelity to EBPs. The rise in global internet access, coupled with COVID, has highlighted opportunities to leverage technology to increase access to care; however, contextual variables, such as network connectivity and user preferences, must be examined and considered when designing and implementing mobile technology interventions and implementation strategies. We describe our approach, which engaged lay counselors and supervisors to design and test a method of optimizing mobile phone supervision. We honored their contextual expertise and engaged them throughout the research process to identify potential barriers and generate solutions to using mobile technology to provide supervision. Ultimately, we co-developed implementation guidelines that were shared and discussed with newly implementing counselors in a brief outreach visit. Results showed significant improvements in mobile phone supervision acceptability and usability. Qualitative results contextualize how the program was effective. Our findings and approach highlight the importance of co-developing flexible and pragmatic implementation strategies to support task shifting and global mental health approaches. 

The manuscript is 4,917 words in length (excluding title pages, abstract, tables, and figures), 3 tables, 3 figures, and 2 supplemental files. This manuscript is not in submission to any other journals. All authors have contributed to and approved this submission. Thank you for your full consideration of this manuscript. I look forward to your review. If you have questions or would like further information, please feel free to contact me at (336) 596-8132 or by email at nst7@uw.edu.

Sincerely,

Noah S. Triplett, MS

PhD Candidate in Clinical Psychology

University of Washington

---

## [Reviewer Report]

*Comments to Author*: Manuscript Review 

Title: Co-developed implementation guidelines to maximize acceptability, feasibility, and usability of mobile phone supervision in Kenya 

These authors present findings on the iterative development and implementation of guidelines focused on facilitating mobile supervision among lay providers administering Trauma-Focused Cognitive Behavioral Therapy (TF-CBT) in Kenya. The framework presented is both innovative and pragmatic and has the potential to optimize supervision-focused implementation in other contexts and with various technologies/modalities. Importantly, the guidelines were developed collaboratively, centering on community participatory principles, emphasizing the expertise of local collaborators. Additionally, the manuscript is well-written, and the methodology and design are very clear. In its current state, this manuscript makes a meaningful contribution. There are a few limitations and points of clarification below that if address may further improve the impact of these findings. 

Major comments/revisions

Method- Implementation Guidelines

• It may be useful for the reader if the authors could expand a bit more on how the educational outreach sessions were conducted. Were these held in person with all supervisors and sequence seven counselors in attendance? Or were sessions conducted via one-on-one contact with supervisors being assigned to hold sessions with a certain number of counselors? Was there a gap between when these sessions were held and when phone supervision began (if this gap was meaningful, it might be helpful to report this)? 

Method- Measures

• It seems plausible that the acceptability, feasibility, and usability constructs may have some conceptual overlap. Would the authors be able to provide correlations between these measures to confirm that these are divergent constructs? 

Method- Analysis 

• Would the authors consider adding some type of effect size for their independent-samples t-test? From the results/ visualization, it appears that the effect of the guidelines may vary by construct. 

• For the qualitative analysis, it would be helpful if the authors included information about the number and qualifications of the coders. Further, it would be informative to know the extent to which the coders were involved in the study and whether/how they were trained on coding procedures. 

Results

• The authors present the means/SDs of acceptability, feasibility, and usability across sequences. However, Figure 3 demonstrates these findings across both sequences and sector. This makes it somewhat difficult to visualize the findings collapsed across sectors. Would the authors be able to either collapse across sectors or present an additional figure that highlights the findings as they are presented in Table 3? 

Discussion 

• A few questions came to mind while reading this manuscript regarding the impact of the guidelines/ outreach sessions: 1) Did these contribute to an increase in number of telephone supervision contacts?; 2) Were there specific strategies that lay counselors/supervisors used more than others?; 3) Did guidelines improve the effectiveness of supervision conducted via phone?; and 4) Did the any of this related to fidelity with implementing the intervention/clinical outcomes. I realize that answering many of these questions is beyond the scope of this manuscript. If possible to address questions 1 and 2, I think those would be interesting to the reader and emphasize the utility of the guidelines/ sessions (above and beyond the utility that is already highlighted!). If you are unable to include these in the manuscript, I may consider a limitation/future direction is that it is unclear how these sessions impact strategies chosen and frequency of these sessions. Questions 3 and 4 may be worth mentioning as future directions if the authors have adequate space. If not, something to ponder on! 

Minor comments/revisions 

Introduction

• On page 5, the authors briefly mention noted challenges of disseminating guidelines. It may be helpful for the reader to briefly expand on what these challenges are (if the authors have sufficient space). 

Method- Participants

• The authors note that there were no exclusion criteria. Was there any inclusion criteria to be a lay counselor for the parent study? If so, would they be able to add that in? 

Figures 

• Figure 3: The x-axis on these figures is difficult to read. The words of the categories listed on the x-axis are overlapping. It’s possible that this is not an issue for others and may be a result of the way the file downloaded on my computer, but I thought it was worth mentioning if the formatting was somehow not converted correctly during the submission process.

---

## [Reviewer Report]

*Comments to Author*: This paper represents an important move forward in providing the use of mobile phone technology supervision of mental health field workers and ultimately better mental health care to those in need. The paper also addresses the need to provide mental health care to formerly underserved and difficult to reach populations in LMIC.

---

## [Reviewer Report]

*Comments to Author*: The study explores a very important subject on co-development of implementation guidelines on mobile phone supervision in Kenya. However, the reviewers have made very relevant observations and recommendations that may improve the quality and content of the manuscript. I invite the authors to respond to comments of the reviewer 1 and resubmit the manuscript.

---

## [Reviewer Report]

April 10, 2023

To Drs. Bass and Chibanda,

Editors-in-Chief 

Cambridge Prisms: Global Mental Health

My co-authors and I are re-submitting the attached article titled, Co-developed implementation guidelines to maximize acceptability, feasibility, and usability of mobile phone supervision in Kenya, for your consideration for publication in Cambridge Prisms: Global Mental Health. Little scholarship has focused how to scale up and sustain task-shifting of evidence-based practices (EBPs) for mental health disorders in lower-resourced settings, including how to sustainably supervise lay counselors. We describe methods and results from an iterative, mixed methods trial that engaged lay counselors and supervisors providing trauma-focused cognitive behavioral therapy (TF-CBT) in Kenya to redesign supervision to most effectively leverage mobile technology. 

We briefly outline the importance of developing scalable and sustainable approaches to supervision—an essential yet resource-intensive implementation strategy for ensuring fidelity to EBPs. The rise in global internet access, coupled with COVID, has highlighted opportunities to leverage technology to increase access to care; however, contextual variables, such as network connectivity and user preferences, must be examined and considered when designing and implementing mobile technology interventions and implementation strategies. We describe our approach, which engaged lay counselors and supervisors to design and test a method of optimizing mobile phone supervision. We honored their contextual expertise and engaged them throughout the research process to identify potential barriers and generate solutions to using mobile technology to provide supervision. Ultimately, we co-developed implementation guidelines that were shared and discussed with newly implementing counselors in a brief outreach visit. Results showed significant improvements in mobile phone supervision acceptability and usability. Qualitative results contextualize how the program was effective. Our findings and approach highlight the importance of co-developing flexible and pragmatic implementation strategies to support task shifting and global mental health approaches. 

The manuscript is 4,991 words in length (excluding title pages, abstract, tables, and figures), 3 tables, 2 figures, and 3 supplemental files. This manuscript is not in submission to any other journals. All authors have contributed to and approved this submission. Thank you for your full consideration of this manuscript. I look forward to your review. If you have questions or would like further information, please feel free to contact me at (336) 596-8132 or by email at nst7@uw.edu.

Sincerely,

Noah S. Triplett, MS

---

## [Reviewer Report]

*Comments to Author*: I appreciate the authors' response to my comments. They have addressed all of my concerns, and I’m happy to recommend this article for publication. Thank you for the opportunity to review this exciting and innovative work!

---

## [Reviewer Report]

*Comments to Author*: This is an excellent paper focused on the urgent need for clinical supervision in low-resource settings. The creative use of mobile phone technology to reach these often overlooked settings is a major movement to level the “playing field” in low-resource settings.

---

## [Reviewer Report]

*Comments to Author*: Thank you for taking into account the feedback provided by the reviewers and incorporating the necessary changes in the revised manuscript. We are pleased to inform you that your manuscript has been accepted for publication.